# Predictive Model of Gait Recovery at One Month after Hip Fracture from a National Cohort of 25,607 Patients: The Hip Fracture Prognosis (HF-Prognosis) Tool

**DOI:** 10.3390/ijerph18073809

**Published:** 2021-04-06

**Authors:** Cristina González de Villaumbrosia, Pilar Sáez López, Isaac Martín de Diego, Carmen Lancho Martín, Marina Cuesta Santa Teresa, Teresa Alarcón, Cristina Ojeda Thies, Rocío Queipo Matas, Juan Ignacio González-Montalvo

**Affiliations:** 1Hospital Universitario Rey Juan Carlos, Universidad Rey Juan Carlos, 28933 Móstoles, Spain; 2Hospital Universitario Fundación Alcorcón, Instituto de Investigación Hospital Universitario La Paz, 28046 Madrid, Spain; pisalop@gmail.com; 3Data Science Lab, Universidad Rey Juan Carlos, 28933 Móstoles, Spain; isaac.martin@urjc.es (I.M.d.D.); carmen.lancho@urjc.es (C.L.M.); marina.cuesta@urjc.es (M.C.S.T.); 4Hospital Universitario La Paz, Instituto de Investigación Hospital Universitario La Paz, 28046 Madrid, Spain; mteresa.alarcon@salud.madrid.org (T.A.); juanignacio.gonzalez@salud.madrid.org (J.I.G.-M.); 5Hospital Universitario 12 De Octubre, 28041 Madrid, Spain; cristina.ojeda@salud.madrid.org; 6Data Science Lab, Universidad Europea de Madrid, 28005 Madrid, Spain; rocio.queipo@universidadeuropea.es

**Keywords:** predictive model, hip fracture, gait recovery

## Abstract

The aim of this study was to develop a predictive model of gait recovery after hip fracture. Data was obtained from a sample of 25,607 patients included in the Spanish National Hip Fracture Registry from 2017 to 2019. The primary outcome was recovery of the baseline level of ambulatory capacity. A logistic regression model was developed using 40% of the sample and the model was validated in the remaining 60% of the sample. The predictors introduced in the model were: age, prefracture gait independence, cognitive impairment, anesthetic risk, fracture type, operative delay, early postoperative mobilization, weight bearing, presence of pressure ulcers and destination at discharge. Five groups of patients or clusters were identified by their predicted probability of recovery, including the most common features of each. A probability threshold of 0.706 in the training set led to an accuracy of the model of 0.64 in the validation set. We present an acceptably accurate predictive model of gait recovery after hip fracture based on the patients’ individual characteristics. This model could aid clinicians to better target programs and interventions in this population.

## 1. Introduction

Hip fractures are highly prevalent in aging societies, with an incidence of over 150 cases annually per 100,000 inhabitants in the general population [1], increasing to 511 cases per year per 100,000 inhabitants over 65 years old [2]. This incidence is expected to rise due to the demographic changes foreseen in the coming decades [3]. These fractures entail high mortality rates (9% the first month after the fracture, 15.5% at 3 months, 26.5% at one year and 36.2% at two years [4,5]), high morbidity and readmission rates (9.3% in the first month [6]). They also affect the patients’ functional status. Even one year after the fracture, approximately 50% of patients have been reported to newly require walking aids, and 90% need help climbing stairs, compared to 21–26% of controls matched for age, sex, comorbidity, and baseline functional status. Functional impairment and increased dependency often become chronic and increase the likelihood of other adverse outcomes such as institutionalization, cognitive impairment, risk of new falls and mortality, worsening the patient’s quality of life and increasing healthcare costs [7,8,9].

The main component of functional recovery is regaining the ability to walk, so a better understanding of the probability that each individual has of recovering baseline ambulatory capacity following a hip fracture could potentially be useful for several reasons: first, clinicians would be able to counsel patients and caregivers on what degree of recovery of the ability to walk can be expected, allowing for better planning of their needs and improved clinical decision making. Second, awareness of the modifiable factors associated with greater deterioration of ambulation could guide clinicians and researchers on interventions to optimize the functional recovery of patients. Advances in surgical techniques, anesthesia, perioperative care, early rehabilitation and multidisciplinary teamwork have improved clinical outcomes in recent years. However, the functional impairment associated with hip fracture remains in need of improvement.

National hip fracture registries have been launched in several countries, allowing for audit of the care process, identification of the appropriateness or deviation from established quality standards and introduction of corrective measures to improve quality of care and efficiency. Often, however, there is insufficient clear data on the functional recovery following hip fracture. While previous studies have analyzed the prevalence of functional impairment and identified several predictive factors, they are generally single-center studies focusing on predictors of functional impairment or cluster analysis [10,11,12]. To our knowledge, no predictive model has been established to estimate the specific probability of each individual’s functional recovery in our setting.

The objective of this study is to develop a predictive model serving as a tool to estimate the individual probability of recovering the previous level of gait independency one month after hip fracture and to build a practical tool applicable in the clinical setting.

## 2. Materials and Methods

### 2.1. Study Sample

We included patients aged 75 years old or older hospitalized for fragility hip fracture between January 2017 and December 2019 at any of the 61 participating hospitals in the Spanish National Hip Fracture Registry (SNHFR). This registry is a previously described initiative carried out voluntarily by a group of clinicians throughout Spain [13]. It is a prospective audit including the variables proposed by the Fragility Fracture Network, endorsed by over 20 regional and national scientific societies. Data is collected during acute hospitalization and one month after the fracture. The SNHFR has established and monitors several standards of quality of care, with the final goal of progressively improving care for patients suffering hip fractures [14].

Patients were included if: they were aged 75 years old or more and admitted to hospital for a fragility hip fracture, included in the SNHFR and provided written informed consent to data collection and analysis (by patients or their legally authorized representatives).

Exclusion criteria were: patients who were non-ambulatory before the fracture or whose walking ability was unknown, and those lost to one-month follow-up due to death, or unknown vital status or walking ability one month after the fracture.

The SNHFR recorded 31,882 cases between 1 January 2017 and 31 December 2019. After excluding 6275 cases, a final sample of 25,607 patients (80.3% of the initial sample) was used, as shown in Figure 1. Another 244 samples were omitted (0.95%) after the missing values analysis.

### 2.2. Data Collection

Study data was collected by over 200 volunteer research clinicians or nurses from hospitals participating in the SNHFR. Baseline data on demographic and clinical variables and self-reported measures of their prefracture functional status was obtained during acute admission from interviewing the individuals or their representatives and through the patients’ medical records. These included age, gender, place of residence (home, nursing home or acute hospitalization), date of arrival to the emergency room, date of surgery (if applicable) and date of discharge to calculate operative delay and total length of stay. Prefracture mobility was collected using the Functional Ambulation Classification Scale (FAC) [15], which scores between 0 (worst) and 5 (best): 0 (no gait at all, or needing the aid of two people), 1 (gait with great assistance of one person), 2 (gait with little help of one person), 3 (gait with supervision of one person), 4 (independent gait on flat surfaces, but needing help to climb stairs) and 5 (independent gait both on flat surfaces and climbing stairs). The target variable was recoded into three categories to facilitate understanding: unable to walk (FAC 0), dependent gait (FAC 1,2,3) and independent gait (FAC 4,5). It is summarized in Table 1. This classification emphasizes the need of another person’s assistance to walk, regardless of the technical aids used. Cognitive status during admission was collected using the Pfeiffer Short Portable Mental State Questionnaire [16] (SPMSQ), defining cognitive impairment as 4 errors or more according to the validated Spanish version [17]. Other data collected was: Fracture type (intracapsular, intertrochanteric or subtrochanteric), anesthetic risk according to the American Society of Anesthesiologists’ (ASA [18]) Physical Status Classification, type of anesthesia (general or spinal), early postoperative mobilization (in the first 24 h after surgery), development of pressure ulcers (grade II or more), involvement of a clinician in addition to the surgical specialist, applying a peripheral nerve block, destination at discharge (private home, nursing home, geriatric rehabilitation unit or other locations such as acute hospitalization or a long stay units), and vital status at discharge. Authorization of weight bearing on the operated limb started to be recorded mid-2018.

Follow-up data was obtained one month after the fracture by contacting the patients by telephone or in person during the follow-up visit. Information regarding vital status and level of ambulation (again using the Functional Ambulation Classification Scale [19]) was collected.

Study protocols were approved by the local institutional review boards of each of the 61 participating centers. A representative in each participating hospital is in charge of data custody and submission in an encrypted format at defined intervals to the registry’s data manager, who is responsible for data cleaning, analysis, and database maintenance.

### 2.3. Outcome Definition

The primary outcome of this study is the recovery of the previous level of walking ability, defined as one of two possible outcomes: for previously independent patients (FAC 4,5), if they were again independent at follow-up (FAC 4,5). For previously dependent patients (FAC 1,2,3), if they were still able to walk at follow-up (FAC > 0). Patients meeting these criteria were defined as “patients recovering ambulation” and those who did not, “patients not recovering ambulation”.

### 2.4. Statistical Analysis

An exploratory analysis was performed first to study variable distributions, the relationship of each variable with the outcome variable, and the presence of missing values. Missing values were treated in the following manner: if the percentage of the missing values was greater than 1%, a new category was created. That applies to the following variables; cognitive impairment (15.9% missing values), ASA (3.7%), weight bearing not allowed (61.2%) and pressure ulcers (1.6%). Regarding weight bearing, the high number of missing data can be explained by the fact that it was a variable that started to be collected after initiating the study. When the percentage of missing values was under 1%, as was the case for the rest of variables, that missing data was deleted. As a consequence, 0.95% of the observations from the entire dataset were eliminated.

Afterwards, in order to reduce overfitting, we split our sample into a training set (20% of the sample), test set (20%) and validation set (60%) [20]. Training and test sets were used in the first phase of the study to develop the predictive model. To assess the model’s accuracy, the developed model was then tested to correctly predict the outcome in a different group of patients (the validation sample).

#### 2.4.1. Training Phase

A descriptive analysis of the variables included in the SNHFR and relationship of each regressive variable with the target variable was carried out as follows: qualitative variables were expressed as counts and percentages, while continuous variables (age, presurgical length of stay, and total length of stay) were described using the median and interquartile range, after observing they did not conform to normal distribution using the Kolmogorov Smirnov test. Baseline characteristics of patients who recovered ambulation were compared with patients not recovering ambulation using the Chi-squared test for categorical variables and the Kruskal-Wallis test for continuous variables. Significance was set with an alpha error of *p* = 0.05. Univariate analysis allowed detecting which variables showed significant differences between groups.

The relationship between the explanatory variables and the target variable “recovery of ambulation” was studied using a logistic regression model, that allowed the likelihood of recovering ambulation to be estimated based on the values observed in the explanatory variables. All variables significantly associated with recovery of ambulation in the univariate analysis were selected and subjected to logistic regression. To select the best logistic regression model, cross-validation and a step-by-step selection of variables in the training set was applied.

Variables with several categories were recoded defining the most frequent category as the reference in the logistic regression model, shown in first place. This was the case for anesthetic risk (ASA III being the most common category), destination at discharge (home) and operative delay (more than 24 h). Some categories were grouped with the reference category, as a significant relation with the target variable was not found in the logistic regression. This was the case for the missing categories regarding pressure ulcers and weight bearing. Intracapsular and intertrochanteric fractures were similarly grouped together.

The probability of recovering ambulation for each individual of the training set through the adjusted regression model was calculated. A calculator using this model called the Hip Fracture Prognosis (HF-prognosis) tool will be included on the SNHFR website (http://rnfc.es/, accessed on 20 March 2021) and/or in a smartphone app.

The optimal threshold for the classification as presence or absence of recovery of ambulation was calculated with the probabilities obtained, optimizing the F1 score 21. This way, a trade-off between precision and recall is achieved [21].

Several cut-off points were defined for the probability of recovering ambulation, creating 5 groups: “very low” (probability less than 0.2), “low” (0.2–0.4), “medium” (0.4–0.6), “high” (0.6–0.8), and “very high” (probability greater than 0.8), allowing us to study the relationship between the predicted probabilities and the observed response variable.

#### 2.4.2. Test Phase

The test set was used to ensure an adequate generalization of the results obtained in the training phase.

#### 2.4.3. Validation Phase

The performance of the adjusted model that is presented in the next section as well as in the probability plot has been obtained using the validation set. All statistical analysis was performed with IBM SPSS Statistics version 27.0 software (IBM, Armonk, NY, USA).

## 3. Results

The baseline characteristics of the overall population and in the groups of patients who do and do not recover ambulation are shown in Table 2. All baseline characteristics had statistically significant differences between the group of patients with and without gait recovery, except gender (*p* value = 0.278), and the evaluation of a clinical doctor in addition to the Traumatology surgeon (*p* value = 0.167).

### 3.1. Training Set; Logistic Regression Model

Figure 2 summarizes the adjusted logistic regression model explaining the target variable “recovery of ambulation” through the explanatory variables. The odds ratios [22] resulting from that model are shown in Table 3.

### 3.2. Performance Measures in Validation

The optimum threshold for the prediction of the presence or absence of recovery of ambulation in patients included in the training set was 0.706. This threshold led to an accuracy of 0.64, a precision of 0.48, a recall of 0.74, a specificity of 0.58 and a F1 score of 0.59 in the validation set.

### 3.3. Groups by Predicted Probability of Recovery

Figure 3 shows the percentage of patients in each of the five groups of predicted probability of recovery, being the most common group the one with high probability of recovery. Figure 4 shows the observed percentages of recovered patients in each of the groups in the validation set. For example, 27.3% of the patients had a very high predicted probability of recovery. Of these, 86.8% recovered. Meanwhile, only 1.8% of the validation sample was classified as patients with a very low probability of recovery, of which only 22.1% managed to recover.

The most common features of each group are summarized in Figure 5, and were the following:“Very low” group (less than 0.2 probability). This group represented 1.8% of the sample. With a mean age of 89.6 years, was characterized by patients admitted for subtrochanteric fractures, in whom postoperative weight bearing was not allowed, who develop pressure ulcers and who are discharged to locations other than their home, nursing homes or geriatric rehabilitation units, such as acute hospitalization or long-stay units.“Low” group (probability 0.2–0.4). This group represented 8.6% of the sample. Patients discharged to nursing homes, suffering subtrochanteric fractures, developing pressure ulcers, and not mobilized on the first postoperative day, characterized this group, that had an average age of 90.2 years. The scarcity of mild systemic disease (ASA = I–II) also stand out.“Medium” group (probability 0.4–0.6). This group represented 23.2% of the sample. With an average age of 88.7 years, was characterized by patients who walked with assistance (FAC 1–3) before the fracture, discharged to nursing homes, and who had cognitive impairment. The scarcity of mild systemic disease (ASA = I–II) was also noteworthy.“High” group (probability 0.6–0.8): This group represented 39.1% of the sample. With an average age of 86.6 years, was characterized by the scarcity of patients discharged to nursing homes and by the predominance of patients in whom weight bearing was allowed and who were sent to geriatric rehabilitation units at discharge.“Very high” group (probability greater than 0.8): This group represented 27.3% of the sample. The mean age of this group was 83 years. It was characterized by the scarcity of patients dependent for walking at baseline, as well as the scarcity of cognitive impairment, subtrochanteric fractures and pressure ulcers. ASA levels = I and II stand out as well as mobilization on the first postoperative day. There is also a predominance of short operative delay (less than 24 h) and discharge back home.

### 3.4. Examples of Hypothetical Patients with Different Probabilities of Recovery

Here we show how the predicted probability of patients’ recovery of ambulation is affected in the validation database when its explanatory variables are modified.

A 79-year-old patient who walks with little assistance (FAC = 2), is cognitively impaired, has mild systemic disease (ASA = II), suffered an intracapsular fracture. Surgery was delayed more than 24 h, the patient was not mobilized on the first postoperative day, but weight bearing was authorized; no pressure ulcers were developed, and the patient was discharged home. The estimated probability of recovery is 0.778. For this same patient:-If weight bearing had not been authorized, probability of recovery would drop to 0.343.-If weight bearing had not been authorized in addition to discharge to a nursing home, the probability of recovery would fall to 0.163.

A 96-year-old patient without cognitive impairment, but requiring continuous support of another person to walk (FAC = 1), who has suffered an intertrochanteric fracture, with an anesthetic risk score of ASA = III, has surgery delayed more than 24 h after admission, and was not mobilized on the first postoperative day. The patient did not develop pressure ulcers and was discharged to a nursing home. The probability of recovering ambulation is 0.506. If this patient:-Had undergone surgery within 24 h after admission, the probability of recovery would be 0.543.-Had been mobilized the first day after surgery, the probability of recovery would be 0.614.-If both situations had occurred, the probability would reach 0.649.-If weight bearing had not been authorized then the probability of recovery would fall to 0.132.

## 4. Discussion

This study proposes a method to estimate the probability of recovering previous ambulation one month after hip fracture. It allows us to stratify hip fracture patients into five risk groups (very low, low, medium, high, and very high probability of recovering ambulation).

We have identified several predictors of recovering ambulation after hip fracture, consistent with most studies. Age is one of the most cited factors, with older patients taking longer to regain their baseline walking ability or at greater risk of not achieving it [23,24,25,26,27,28]. Gender did not significantly affect ambulatory recovery in our study, nor in previous ones [29,30,31]. Our results, however differed from others regarding previous ambulatory dependence; while in other studies it seems to be a risk factor for not recovering ambulation [23,26,27] as also observed in the univariate analysis in our study, it proved to be a factor in favor of recovery in our regression model. This could be explained by the wide definition we have made of recovery of ambulation, despite it being similar to that of other authors. For example, in our study, a patient who walked regularly with supervision (FAC 3), may be more likely to recover as the margin up to the category of FAC 1 is wider, while for a patient with a baseline FAC 4 ambulation level, deteriorating to FAC 3 at one month was considered “not recovered”.

Other risk factors of not recovering ambulation identified in our study, as well as in previous ones, are comorbidity [25,27,32], cognitive impairment [24,25,26], subtrochanteric fractures [24,26], not permitting weight-bearing [33,34], and late postoperative mobilization [35]. Operative delay has widely been studied as a risk factor for mortality, but there are few studies focusing on its relation with functional recovery [36,37]. In our study, an operative delay of more than 24 h was related to a higher risk of not recovering ambulation in the univariate analysis, but lost its significance in the multivariate analysis. This could be due to the relatively low proportion of patients treated in less than 24 h, inferior to that reported by other audits [13]. Analysis of other thresholds for delay and the combined effect of early surgery and early postoperative mobilization would be of interest. Pressure ulcers have been associated with longer surgical delay and increased mortality and morbidity [38] but their relation with recovery of ambulation has been studied less. We observed that they were associated with a lower probability of recovering ambulation, perhaps acting as an effect as well as a cause of reduced mobilization during hospitalization. The discharge destination showing the greatest likelihood of ambulatory recovery was home, followed by geriatric rehabilitation units; nursing homes and other locations such as acute hospitalization were associated with the lowest probability of recovering ambulation. The lack of recovery of walking ability could be a cause rather than consequence of the discharge destination. For example, a patient who recovers quickly and adequately during acute admission is more likely to return home, while a patient who does not regain ambulation—perhaps due to numerous intercurrent medical complications—is more likely to be transferred to a geriatric rehabilitation unit, and has still not recovered after a month. Finally, a patient who does not regain ambulation during acute admission and has little chance of doing so even in a rehabilitation facility is more likely to be sent to a nursing home. The appearance of medical complications, which could explain the worse functional results observed compared to those expected in some patients, is not included in the SNHFR and could therefore not be controlled for in the multivariate analysis.

Several factors such as the anesthetic risk are non-modifiable. Others, however, are modifiable and have an even greater impact on the likelihood of recovery, such as the permission to bear weight. With an Odds Ratio of 0.149, the risk of recovery is 1/0.149 = 6.71 times higher in patients in whom weight bearing is permitted versus patients who are not allowed to bear weight. The modifiable factors found were time to surgery, early mobilization, weight-bearing status, and the development of pressure ulcers. Although the best discharge destination in our study was home, it is important to have the possibility to discharge patients to geriatric rehabilitation units in order to minimize the loss of ambulatory independency in complex patients. It is likely that a longer follow-up is needed to fully appreciate their effect.

Predictors of ambulatory recovery were similar to other studies focusing on tools that calculate the individual probability of recovering ambulation. Other factors not included in our model are gender, body mass index, polypharmacy, type of surgery performed, pre-fracture Barthel Index, and postoperative complications [32,39,40,41]. One of the studies cited has the advantage of using a longer (one year) follow-up [41]. Some developed a regression model similar to the one presented here [32,41], while others have reported a score instead [32,39]. All of them have been validated using less than 500 patients, and none of them have been validated in our setting. The most similar predictive model is the one developed by Kim et al., who retrospectively reviewed patients aged 60 and older with hip fractures and developed a predictive model of ambulation at one month postoperatively, which included age, gender, prefracture ambulation and the generic term of “combined medical diseases”, defined as diseases which could affect ambulatory capacity. Their model showed an accuracy of 0.704 but did not include any modifiable variables on which to act [32].

One of the strengths of our study is the large number of patients included, offering enough statistical power to be able to study multiple predictive factors of gait recovery. The casemix reported by the SNHFR is similar to that in other studies from Spain [42,43,44,45,46,47], and it can be inferred it is representative of the population of hip fractures. The possibility of calculating the individual probability of recovering ambulation using a simple web based calculator makes this study particularly interesting.

We are aware our study has several weaknesses. First, comorbidity was adjusted using to the ASA score instead of a specific score for comorbidity. The ASA score is however collected universally through hospitals and recommended in the Fragility Fracture Network dataset for hip fracture audit. Second, possible confounding factors like the presence of medical complications during admission was not recorded, as previously mentioned. Data on the rehabilitation and mobilization carried out during admission—not only on the first postoperative day—would also be interesting to take into account. Third, this is an observational study; the theoretical contribution of each factor to ambulatory recovery can be examined, and hypotheses can be formulated, but the effect of individual interventions on each of the prognostic factors cannot be quantified, nor can causality be determined. We are limited to showing statistically significant associations. Fourth, ambulatory capacity was measured using the FAC scale, which relies mainly on the responses given by the patient and/or family during admission. This is less accurate than measures such as the Timed up and go [48], SPPB [49] tests. In the setting of an acute hip fracture, it would be impossible to obtain preoperative baseline measurements for these tests. Fifth, more than 200 professionals are involved in data collection, which may increase the risk of classification errors and variability, especially if the one-month follow-up information was obtained by telephone instead of in person. Finally, follow-up is carried out at one month. Thirty days is a very short period for recovery compared with other studies and registries with longer follow-up periods (usually 3 or 6 months), which more accurately determine the level of long-term functional recovery. Again, our registry follows the Fragility Fracture Network’s Minimum Common Dataset, which recommends 30-day follow-up. In our opinion, predicting short-term recovery is also interesting. Few studies that provide information on the ability to walk at one month, and the ability to walk at 30 days is an important factor in planning short-term patient care. It also could potentially be a predictor of recovery at 3 or 6 months or even 1 year.

In summary, hip fracture commonly leads to dependence in older patients that frequently persists one month after the fracture. The ambulatory capacity of patients with a lower functional reserve is more likely compared to those who do not deteriorate. The former group is more likely to be older, be admitted from nursing homes, have dementia or significant comorbidity, and therefore has a more unfavorable baseline situation. They are less likely to receive peripheral nerve blocks, to be operated on in less than 24 h or be mobilized on the first postoperative day, and they are referred to rehabilitation units less frequently. In order to maximize the likelihood of recovering the ability to walk, our resources should be focused on optimizing these modifiable factors.

A tool that identifies the probability of each patient to recover pre-fracture walking independence following hip fracture surgery is useful for aiding clinicians and health care administrators to develop targeted programs and interventions. Healthcare administrators could also use it to estimate the need of post-acute care facilities adapted to patients’ needs. This aid is also useful to counsel patients and caregivers on the functional prognosis following hip fracture, so they can plan the social support they may need.

Further studies are, of course, needed to validate this tool for different populations and settings. More complex machine learning models rather than logistic regression models could help us to improve this model in the near future.

## 5. Conclusions

The aim of this study was the development of a predictive model that aid clinicians to calculate the probability for patient of recovering the baseline level of independent or assisted ambulation one month after hip fracture. This is important, as the ability to walk is the main component of functional recovery after a hip fracture.

In this study of 25,607 Spanish hip fracture patients, 16,839 (65.8%) recovered prefracture ambulatory capacity at one month. Non-modifiable risk factors independently associated with worse recovery were cognitive impairment, elevated anesthetic risk, and a suffering subtrochanteric fractures. The modifiable factors affecting the probability of regaining prefracture ambulation were performing surgery in less than 24 h, early mobilization, allowance of weight-bearing, avoiding pressure ulcers and discharging the patient home.

We have successfully developed a predictive model that included all these factors, based on logistic regression. Our model has shown similar accuracy to prior studies by other authors. This predictive model estimates the individual probability of recovery (ranging from 0 to 100%), and we have defined 5 groups or clusters of patients depending on this probability. Patients with a very low and low probability of recovery are usually older, have suffered subtrochanteric fractures, are not allowed to bear weight, and are commonly discharged to nursing homes. On the other hand, patients with high and very high probabilities of recovery are younger, without cognitive impairment, present with intracapsular or intertrochanteric fractures, and walked independently before the fracture; they also have a shorter surgical delay and are likely to be sent home at discharge. In other words, those with the least probability of recovery start off from a worse baseline situation, but also managed more poorly, with worse performance indicators during admission, such as greater surgical delay or not allowing weight bearing. On the opposite end of the spectrum, patients most likely to recover have a better prefracture status and are better managed, fulfilling key performance indicators.

These findings can be used to aid risk stratification in this population to support informed treatment decisions and to aid conversations regarding goals of care. Future work on this topic could refine the precision of the model using more complex machine learning models, and longer follow-up time.

## Figures and Tables

**Figure 1 ijerph-18-03809-f001:**
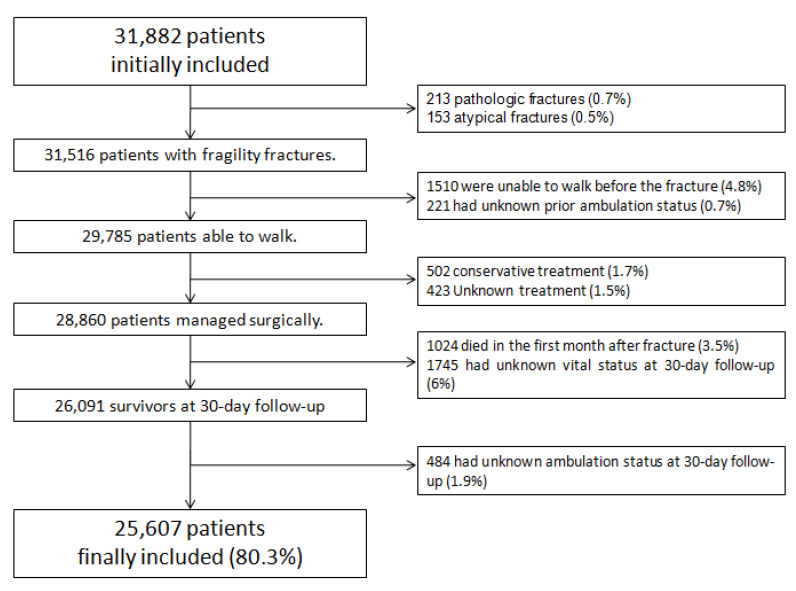
Inclusion and exclusion criteria.

**Figure 2 ijerph-18-03809-f002:**
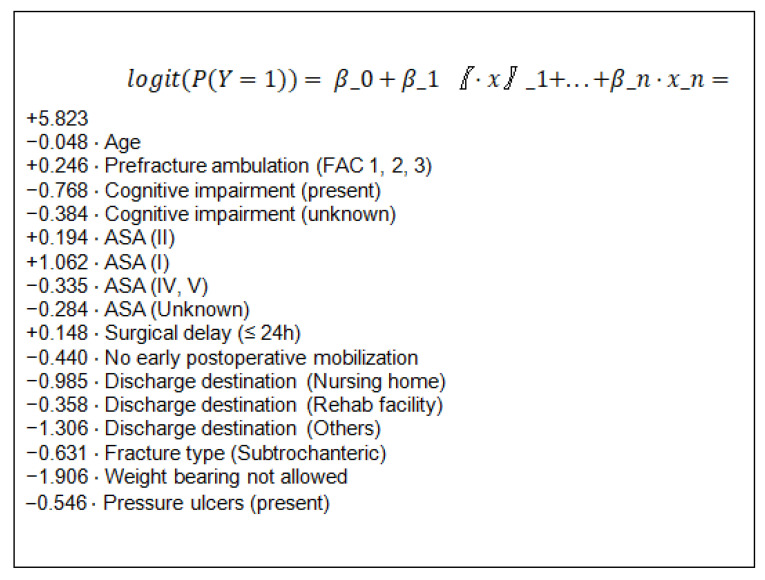
Equation of the logistic regression model. Note that in the model some variables appear several times; this is because each time it refers to one of the variable’s categories. Thus, for example, the ASA variable has 5 categories: reference (ASA III), I, II, IV–V and unknown. ASA III level does not appear in the formula because this is the reference level. If a patient has ASA I, in the model it will be translated as ASA I = 1 and the rest of ASA levels = 0. If a patient has ASA II, it will be ASA II = 1 and the rest of the ASA levels will be = 0.

**Figure 3 ijerph-18-03809-f003:**
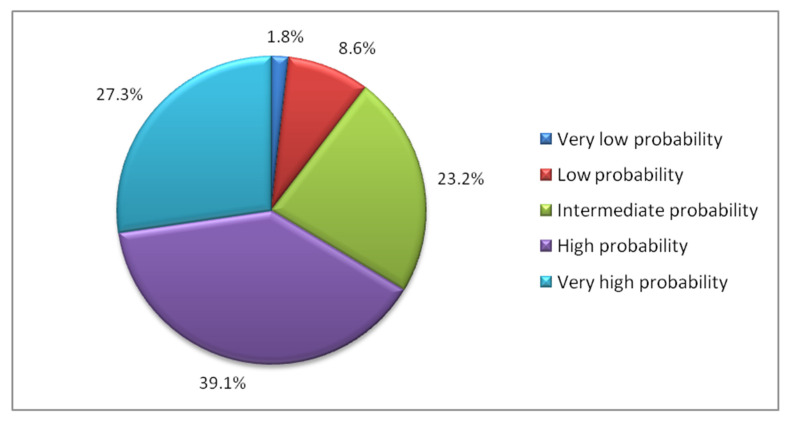
Distribution of the sample according to the groups of predicted probability of recovery.

**Figure 4 ijerph-18-03809-f004:**
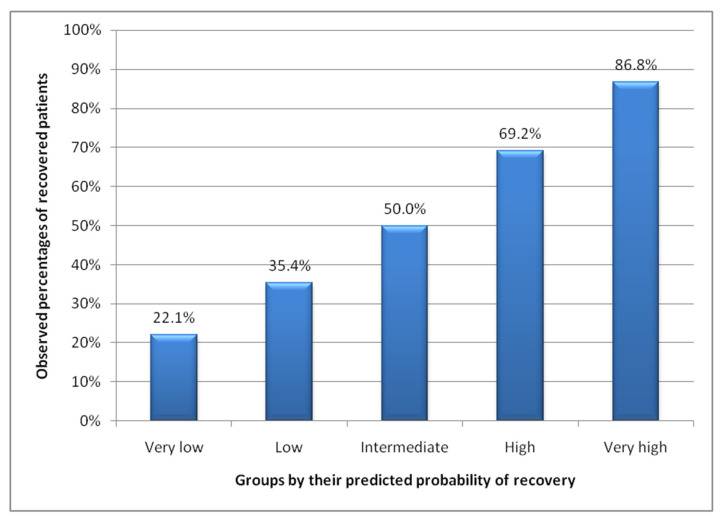
Rate of recovered patients within each group by their predicted probability of recovery.

**Figure 5 ijerph-18-03809-f005:**
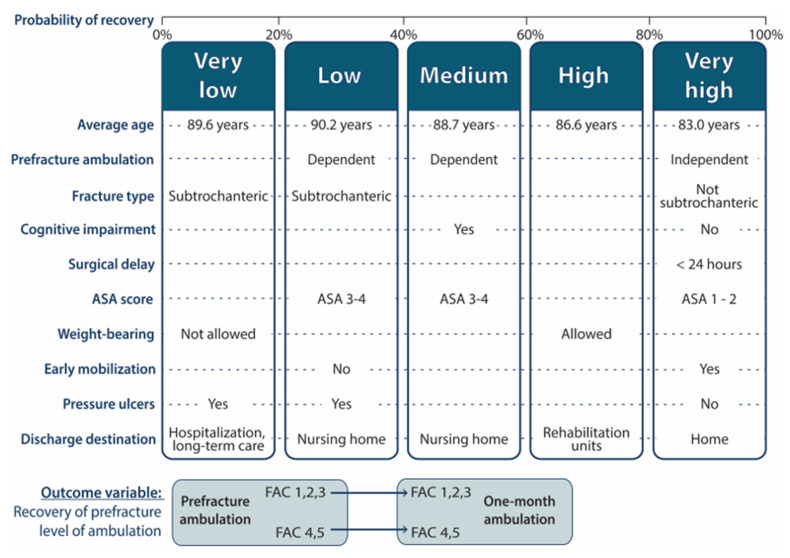
Clusters predicting the probability of functional recovery (very low; 0–20%; low, 20–40%; medium, 40–60%; high, 60–80%; very high, 80–100%), The boxes show the average age of the patients include in each cluster, as well as the most relevant features for the variables included in the model. Dashed lines represent transitions in which the difference between one cluster and the next were not relevant for the variable. Below: definition of the outcome variable as recovery of prefracture ambulation at one month, depending on FAC categories (1,2,3 vs. 4,5).

**Table 1 ijerph-18-03809-t001:** Functional ambulation classification scale.

Functional Ambulation Category	Description
5	Independent, all surfaces
4	Independent, level surfaces only
3	Dependent for supervision
2	Dependent for physical assistance—level I (light touch)
1	Dependent for physical assistance—level II (support body weight)
0	Nonambulator

**Table 2 ijerph-18-03809-t002:** Baseline characteristics of the global sample and according to the presence or absence of gait recovery. Univariate analysis.

		All Patients *n* = 25,607	Patients Recovering Ambulation *n* = 16,839 (65.8)	Patients Not Recovering Ambulation *n* = 8768 (34.2)	*p* Value
Age (years) Median ± IQR		87 (83–90)	86 (82–90)	88 (84–91)	<0.001
Gender (Female) *n* (%)		19,753 (77.1)	13,024 (77.4)	6729 (76.8)	0.278
Place of residence: nursing home *n* (%)		5404 (21.1)	2585 (15.4)	2819 (32.2)	<0.001
Prefracture ambulation *n* (%)	FAC 1	1073 (4.2)	514 (3.1)	559 (6.4)	<0.001
	FAC 2	1072 (4.2)	659 (3.9)	6413 (4.7)	
	FAC 3	908 (3.5)	616 (3.7)	292 (3.3)	
	FAC 4	7588 (29.6)	3662 (21.7)	3926 (44.8)	
	FAC 5	14,966 (58.4)	11,388 (67.6)	3578 (40.8)	
Cognitive impairment *n* (%)		8701 (40.4)	4589 (32.1)	4112 (57.0)	<0.001
ASA *n* (%)	I	295 (1.2)	253 (1.6)	42 (0.5)	<0.001
	II	7063 (28.6)	5224 (32.2)	1839 (21.8)	
	III	15,032 (61)	9481 (58.4)	5551 (65.9)	
	IV–V	2266 (9.2)	1275 (7.9)	991 (11.8)	
Fracture type *n* (%)	Intracapsular	10,005 (39.3)	7266 (43.4)	2739 (31.5)	<0.001
	Intertrochanteric	13,496 (53.1)	8490 (50.8)	5006 (57.5)	
	Subtrochanteric	1925 (7.6)	968 (5.8)	957 (11.0)	
Spinal anesthesia *n* (%)		23,948 (93.9)	15,805 (94.2)	8143 (93.5)	0.036
Peripheral nerve block *n* (%)		3534 (16.5)	2472 (18.0)	1062 (13.8)	<0.001
Time to surgery (hours) Median ± IQR		50.8 (26.1–90)	49.4 (25.1–89)	54,5 (28.2–91.6)	<0.001
Surgery in the first 24 h *n* (%)		5788 (22.6)	3966 (23.6)	1822 (20.8)	<0.001
Mobilization on the first postoperative day *n* (%)		17,685 (69.1)	12,354 (73.4)	5331 (60.9)	<0.001
Weight bearing not permitted *n* (%)		789 (7.9)	230 (3.6)	559 (15.8)	<0.001
Pressure ulcers *n* (%)		1233 (4.9)	592 (3.6)	641 (7.5)	<0.001
Clinician in addition to surgeon. *n* (%)		24,615 (96.2)	16,206 (96.3)	8409 (95.9)	0.167
Discharge destination *n* (%)	Home	11,345 (44.3)	8743 (51.9)	2602 (29.7)	<0.001
	Nursing home	7910 (30.9)	3928 (23.3)	3982 (45.4)	
	Geriatric rehabilitation unit	5893 (23.0)	3985 (23.7)	1908 (21.8)	
	Other	450 (1.8)	178 (1.1)	272 (3.1)	
Length of stay (days) Median ± IQR		8.9 (6.6–12.1)	8.7 (6.5–11.8)	9.2 (6.8–12.9)	<0.001

Abbreviations: IQR = interquartile range. FAC = Functional Ambulation Clasification. ASA = American Society of Anesthesiologists’ Physical Status Classification.

**Table 3 ijerph-18-03809-t003:** Results of the logistic regression model. Odds ratios of the explanatory variables, with the target variable “recovery of ambulation”.

		Odds Ratio	95% Confidence Interval	
			Lower Margin	Higher Margin
Age		0.953	0.942	0.964
Prefracture ambulation	Dependent (FAC 1–3)	1.279	1.055	1.550
Cognitive impairment	Present	0.464	0.401	0.537
	Unknown	0.681	0.566	0.820
ASA	I	2.891	1.273	6.565
	II	1.214	1.043	1.413
	IV–V	0.715	0.573	0.892
	Unknown	0.753	0.539	1.052
Type of fracture; Subtrochanteric fracture		0.532	0.422	0.671
Surgical delay ≤24 h.		1.159	0.993	1.352
Postoperative mobilization >24 h		0.644	0.562	0.737
Weight bearing Not allowed		0.149	0.098	0.224
Pressure ulcers Present		0.579	0.443	0.758
Discharge destination	Nursing home	0.373	0.321	0.434
	Geriatric rehabilitation unit	0.699	0.592	0.826
	Other	0.271	0.173	0.425

Abbreviations: FAC = Functional Ambulation Clasification. ASA = American Society of Anesthesiologists’ Physical Status Classification. Reference categories are: Prefracture ambulation; Independent (FAC 4–5). Cognitive impairment; absent. Anaesthetic risk; ASA III. Type of fracture; intracapsular or intertrochanteric. Surgical delay; >24 h. Postoperative mobilization; ≤24 h. Weight bearing; allowed or unknown. Development of pressure ulcers; absent or unknown. Discharge destination; home.

## Data Availability

The data obtained from the SNHFR is anonymized and guarded by the data manager. They are available to the participants in the registry for the preparation of reports with the intention of anonymously comparing the results between centers, and for the development of studies, under the supervision of the organizing board of the Registry. The data is not accessible on any freely accessible web page.

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
