# Peer review of "Predictive Model of Gait Recovery at One Month after Hip Fracture from a National Cohort of 25,607 Patients: The Hip Fracture Prognosis (HF-Prognosis) Tool"

_ijerph, 2021, doi:10.3390/ijerph18073809_

Round 1

Reviewer 1 Report

Your study aimed to develop a predictive model serving as a tool to es-timate the individual probability of recovering the previous level of gait independency one month after hip fracture and to build a practical tool applicable in the clinical setting.

Major revision:

: I think that your selected variables are known, and your prediction model is standard. Is your strength only the large number of patients?

Please write your strength points and clinical implication clearly on the based of your results.

: "To our knowledge, no predictive model has been established to estimate the specific probability of each individual's functional recovery in our setting". In light of the literature review, this statement is overstated. I think that there are a few studies.

Minor revision:

: Figure 5: "−−" is confusing. Change the better description.

: "A probability threshold of 0.706 in the training set" is an acceptably accurate predictive model?

  Please compare your results with the prediction model for hip fractures in previous studies, and descript the data's difference with the table if possible.

: please read instructions to the author. Revise how to describe citation in the manuscript.

Author Response

Your study aimed to develop a predictive model serving as a tool to estimate the individual probability of recovering the previous level of gait independency one month after hip fracture and to build a practical tool applicable in the clinical setting.

Major revision:

  1. I think that your selected variables are known, and your prediction model is standard. Is your strength only the large number of patients?

Authors’ reply: We agree with the reviewer that our prediction model is standard. Advanced models such as ensembler-based models could be considered in the future.

  1. Please write your strength points and clinical implication clearly on the based of your results.

Authors’ reply: One of the strengths of our study is the large number of patients included, allowing enough statistical power and representation of the Spanish real population of hip fracture patients. Another advantage is that it does not only study the risk factors associated with functional recovery, but also calculates the probability of recovery for each individual patient.

Another advantage is the applicability of the tool. 300 professionals from 80 hospitals participate in the SNHFR and continuously collect the variables contemplated by this prediction model. This prediction model will be available on the web and will be used by participants to inform about the predicted probability of recovery and to aid in decision-making.

  1. "To our knowledge, no predictive model has been established to estimate the specific probability of each individual's functional recovery in our setting". In light of the literature review, this statement is overstated. I think that there are a few studies.

Authors’ reply: We agree with the reviewer that there are more studies, but non of them in our setting (Spain). Belleli et al described a similar one but in Italy, Zuckermann and Lee in USA, Hirose in Japan, Kim in Korea, etc…

Minor revision:

  1. Figure 5: "−−" is confusing. Change the better description.

Authors’ reply: Clusters predicting the probability of functional recovery (very low; 0-20%; low, 20-40%; medium, 40-60%; high, 60-80%; very high, 80-100%), The boxes show the average age of the patients include in each cluster, as well as the most relevant features for the variables included in the model. Dashed lines represent transitions in which the difference between one cluster and the next were not relevant for the variable. Below: definition of the outcome variable as recovery of prefracture ambulation at one month, depending on FAC categories (1,2,3 vs. 4,5)

  1. "A probability threshold of 0.706 in the training set" is an acceptably accurate predictive model?

Authors’ reply: The complete sentence in the manuscript is “A probability threshold of 0.706 in the training set led to an accuracy of the model of 0.64 in the validation set.”

Thus, the value of 0.706 refers to the reference value for an individual probability. In other words, if the specific probability of an individual's functional recovery is lower than 0.706, the prediction is “no recovery”. In all other cases, the prediction given by the model would be “recovery”.

Furthermore, in our study we propose a categorization of the probabilities given by the model obtaining 5 groups by predicted probability of recovery (“very low”, “low”, “medium”, “high”, and “very high”). Thus, 4 probability thresholds are needed: 0.2, 0.4, 0.6 and 0.8.

The accuracy (0.64) observed is similar to those observed with other commonly used scoring systems, such as the ASA anaesthetic risk score and Charlson Comorbidity Index for for 30-day mortality following hip fracture. (Marufu TC, Mannings A, Moppett IK. Risk scoring models for predicting peri-operative morbidity and mortality in people with fragility hip fractures: Qualitative systematic review. Injury. 2015 Dec;46(12):2325-34. doi: 10.1016/j.injury.2015.10.025.). And also similar to other predictive models of functional recovery (Hirose J, Ide J, Yakushiji T, et al. Prediction of Postoperative Ambulatory Status 1 Year After Hip Fracture Surgery. Archives of Physical Medicine and Rehabilitation. 2010;91(1):67-72. doi:10.1016/j.apmr.2009.09.018; Bellelli G, Noale M, Guerini F, et al. A prognostic model predicting recovery of walking independence of elderly patients after hip-fracture surgery. An experiment in a rehabilitation unit in Northern Italy. Osteoporos Int. 2012;23(8):2189-2200. doi:10.1007/s00198-011-1849-x)

  1. Please compare your results with the prediction model for hip fractures in previous studies, and descript the data's difference with the table if possible.

Authors’ reply: It is summarized in this part of the discussion;

“Predictors of gait recovery are similar among other studies focusing on tools to calculate the individual probability of recovering ambulation. Other factors not included in our model are gender, body mass index, polypharmacy, type of surgical intervention, pre-fracture Barthel Index, and postoperative complications [32,39–41]. One of these studies has the advantage of using a longer (one year) follow-up 41. Others have developed a regression model similar to the one presented [32,41], while others have reported a score instead [32,39]. All of them have been validated in less than 500 patients, and none of them have been validated in our setting”

And we have added the following sentence, thank you very much for the suggestion: “The most similar predictive model is the one developed by Kim et al, that retrospectively reviewed patients aged 60 and older with hip fractures and developed a predictive model of ambulation at one month postoperatively, which included age, gender, prefracture ambulation and the generic term of “combined medical diseases”, defined as diseases which could affect ambulatory capacity. Their model showed an accuracy of 0.704, but did not include any modifiable variables on which to act.

  1. Please read instructions to the author. Revise how to describe citation in the manuscript.

Authors’ reply: Thank you, we have added brackets to the citations.

Reviewer 2 Report

In Abstract, this sentence, “The primary outcome was recovery of the previous level of gait independence.” Is confusing. Please, consider rewording it.

Why did decide to use 40% of the sample to create the model and 60 to validate it? Is that a common practice?

You claim your model is acceptably accurate. Based on what do you Mae this assumption that it’s acceptable?

Your introduction is too short, please expand it with more literature review.

What made you decide to include only patients of the age 75+?

You state that the 1-month follow up information was collected over the phone only. What confidence do you have that the data collected that way wasn’t too subjective?

Did you, or are you going to, also test your model for predicting recovery at more months after the surgery? 

Author Response

  1. In Abstract, this sentence, “The primary outcome was recovery of the previous level of gait independence.” Is confusing. Please, consider rewording it.

 Authors’ reply: We have rephrased the sentence: “The primary outcome was recovery of the baseline level of ambulatory capacity”

  1. Why did decide to use 40% of the sample to create the model and 60 to validate it? Is that a common practice?

Authors’ reply: We decided to use 40% of the sample for training, and 60% for validation. As is commonly chosen in the Machine Learning area. It is custom to split the data set into two different sets: training and validation. The proportion of data in each set strongly depends on the problem. However, 40-60 is a common choice.

  1. You claim your model is acceptably accurate. Based on what do you Mae this assumption that it’s acceptable?

 Authors’ reply: See our reply to reviewer 1, points 5 and 6.

  1. Your introduction is too short, please expand it with more literature review.

Authors’ reply: We tried to limit the extension of our manuscript but have expanded the literature review somewhat in the discussion.

  1. What made you decide to include only patients of the age 75+?

Authors’ reply: This is the age group commonly receiving co-managed orthogeriatric care in Spain, and the threshold for inclusion in the Spanish National Hip Fracture Registry, the dataset from which the data for this information was procured. Over 60% of hip fractures occur in patients aged 75 years and older, and the greatest controversy regarding functional recovery lies precisely in this older, more fragile, age group. Furthermore, the case volume in this age group is expected to experience the largest increase, due to population aging, showing a “right shoft” in hip fracture incidence.

Kannus P, Parkkari J, Sievänen H, Heinonen A, Vuori I, Järvinen M. Epidemiology of hip fractures. Bone. 1996 Jan;18(1 Suppl):57S-63S. doi: 10.1016/8756-3282(95)00381-9.

Serra JA, Garrido G, Vidán M, Marañón E, Brañas F, Ortiz J. Epidemiología de la fractura de cadera en ancianos en España [Epidemiology of hip fractures in the elderly in Spain]. An Med Interna. 2002 Aug;19(8):389-95.

Zhang C, Feng J, Wang S, Gao P, Xu L, Zhu J, Jia J, Liu L, Liu G, Wang J, Zhan S, Song C. Incidence of and trends in hip fracture among adults in urban China: A nationwide retrospective cohort study. PLoS Med. 2020 Aug 6;17(8):e1003180. doi: 10.1371/journal.pmed.1003180.

Bergström U, Jonsson H, Gustafson Y, Pettersson U, Stenlund H, Svensson O. The hip fracture incidence curve is shifting to the right. Acta Orthop. 2009 Oct;80(5):520-4. doi: 10.3109/17453670903278282.

  1. You state that the 1-month follow up information was collected over the phone only. What confidence do you have that the data collected that way wasn’t too subjective?

Authors’ reply: we are sorry for not having worded this aspect sufficiently clearly, Follow-up was performed by participating clinicians through in-clinic visits, and by telephone when in-person follow-up was not possible. This is standard for nearly all national registries, who use a combination of in-person follow-up as well as telephone interviews and even collection of information by mail, as for example in the UK’s national hip fracture database (https://nhfd.co.uk/20/hipfractureR.nsf/vwContent/FAQs) , which currently is the most important hip fracture database worldwide.

That was we tried to mean with the sentence “Follow-up data was obtained one month after the fracture by contacting the patients by telephone or during the follow-up visit”

  1. Did you, or are you going to, also test your model for predicting recovery at more months after the surgery?

Authors’ reply: Follow-up is limited to 30 days for the Spanish National Hip Fracture Registry, following the recommendations given by the Fragility Fracture Network in their Minimum Common Dataset, and by the SHAFE project (Standardization of Hip Fracture Audit). As our registry gains experience in data collection, expanding to 120-day follow-up as occurs in many other national audits is planned, and some centers are currently piloting the experience. It would definitely be interesting to test our model for prediction at 120 days. However, loss of follow-up at 120 days is greater, particularly among the frailest patients and those with cognitive impairment.

Reviewer 3 Report

The topic is interesting and the study is correctly designed.

However, it requires a deep English check bu an English native speaker.

Please, detail Functional Ambulation Classification in a table. This might help the reader to follow.

I would put "variables finally selected" in table rather than in paragraphs

This paper may help in predicting outcome in patients with hip fracture.

Author Response

REVIEWER 3

The topic is interesting and the study is correctly designed.

However, it requires a deep English check bu an English native speaker.

Authors’ reply: We thank the reviewer for the thorough review. A native English speaker experienced in scientific writing has reviewed the entire manuscript, but we have done some changes somehow.

Please, detail Functional Ambulation Classification in a table. This might help the reader to follow.

Authors’ reply: We have added the Functional Ambulation Classification in a table.

Functional ambulation category

Description

5

Independent, all surfaces

4

Independent, level surfaces only

3

Dependent for supervision

2

Dependent for physical assistance— level I (light touch)

1

Dependent for physical assistance— level II (support body weight)

0

Nonambulator

Functional ambulation category, as described by Holden et al (Holden MK, Gill KM, Magliozzi MR. Gait assessment for neurologically impaired patients. Standards for outcome assessment. Phys Ther. 1986 Oct;66(10):1530-9. doi: 10.1093/ptj/66.10.1530)

I would put "variables finally selected" in table rather than in paragraphs

Authors’ reply: We included the variables finally selected in figure 2 and eliminated them from the text, for easier reading.

This paper may help in predicting outcome in patients with hip fracture.

Authors’ reply: Thank you, we agree it could be an interesting tool to predict functional outcome.

Reviewer 4 Report

  1. The paper deals with a very important topic of recovery of walking ability in patients after hip fracture. The advantage of the work is very large research material
  2. It has not been described what type of hip fracture surgery was performed because that seems to be very  important. It was not specified whether other complications, apart from pressure ulcers, such as infections of the respiratory tract or urinary tract or thromboembolic complications, had an impact on the recovery of walking ability.
  3. References cited in the text are written without parentheses, therefore not readable.
  4. Conclusions should be clearer and include more practical information on how to put the results into practice

Author Response

REVIEWER 4

  1. The paper deals with a very important topic of recovery of walking ability in patients after hip fracture. The advantage of the work is very large research material

Authors’ reply: We wish to thank the reviewer; we believe large audit-based datasets are a useful tool for machine learning and prediction models.

  1. It has not been described what type of hip fracture surgery was performed because that seems to be very It was not specified whether other complications, apart from pressure ulcers, such as infections of the respiratory tract or urinary tract or thromboembolic complications, had an impact on the recovery of walking ability.

Authors’ reply: We have only collected the complications recommended in the by the Fragility Fracture Network in their Minimum Common Dataset, and by the SHAFE project (Standardization of Hip Fracture Audit). The overwhelming majority of other international hip fracture audits do not include the complications mentioned by the reviewer in their datasets, though we do agree they are relevant for patient outcomes.

The type of hip fracture surgery performed depends mainly on the type of fracture suffered (i.e. arthroplasty is much more common for intracapsular fractures, vs. Internal fixation for intertrochanteric fractures). As fracture type was included in our model, and addition of the type of surgery did not increase accuracy

  1. References cited in the text are written without parentheses, therefore not readable.

Authors’ reply: Thank you, we have added brackets to the citations

  1. Conclusions should be clearer and include more practical information on how to put the results into practice

Authors’ reply: Thank you for the suggestion, we have expanded and clarified the information as much as we have been able.